# Associations of Nutritional, Environmental, and Metabolic Biomarkers with Diabetes-Related Mortality in U.S. Adults: The Third National Health and Nutrition Examination Surveys between 1988–1994 and 2016

**DOI:** 10.3390/nu14132629

**Published:** 2022-06-24

**Authors:** Xi Zhang, Shirin Ardeshirrouhanifard, Jing Li, Mingyue Li, Hongji Dai, Yiqing Song

**Affiliations:** 1Clinical Research Unit, Department of Pediatrics, Xin Hua Hospital, Shanghai Jiao Tong University School of Medicine, Shanghai 200092, China; zhangxi@xinhuamed.com.cn; 2Department of Epidemiology, Richard M. Fairbanks School of Public Health, Indiana University, Indianapolis, IN 46202, USA; shirin.rouhani@gmail.com (S.A.); ml57@iu.edu (M.L.); 3Department of Biostatistics, Richard M. Fairbanks School of Public Health, Indiana University, Indianapolis, IN 46202, USA; jing.li.312@gmail.com; 4Department of Epidemiology and Biostatistics, Tianjin Medical University Cancer Institute and Hospital, National Clinical Research Center for Cancer, Tianjin Key Laboratory of Cancer Prevention and Therapy, Tianjin’s Clinical Research Center for Cancer, Key Laboratory of Molecular Cancer Epidemiology of Tianjin, Tianjin 300060, China

**Keywords:** diabetes, nutritional biomarker, environmental biomarker, metabolic biomarkers, all-cause mortality

## Abstract

Background: Nutritional, environmental, and metabolic status may play a role in affecting the progression and prognosis of type 2 diabetes. However, results in identifying prognostic biomarkers among diabetic patients have been inconsistent and inconclusive. We aimed to evaluate the associations of nutritional, environmental, and metabolic status with disease progression and prognosis among diabetic patients. Methods: In a nationally representative sample in the NHANES III (The Third National Health and Nutrition Examination Survey, 1988–1994), we analyzed available data on 44 biomarkers among 2113 diabetic patients aged 20 to 90 years (mean age: 58.2 years) with mortality data followed up through 2016. A panel of 44 biomarkers from blood and urine specimens available from NHANES III were included in this study and the main outcomes as well as the measures are mortalities from all-causes. We performed weighted logistic regression analyses after controlling potential confounders. To assess incremental prognostic values of promising biomarkers beyond traditional risk factors, we compared c-statistics of the adjusted models with and without biomarkers, separately. Results: In total, 1387 (65.2%) deaths were documented between 1988 and 2016. We observed an increased risk of all-cause mortality associated with higher levels of serum C-reactive protein (*p* for trend = 0.0004), thyroid stimulating hormone (*p* for trend = 0.04), lactate dehydrogenase (*p* for trend = 0.02), gamma glutamyl transferase (*p* for trend = 0.02), and plasma fibrinogen (*p* for trend = 0.03), and urine albumin (*p* for trend < 0.0001). In contrast, higher levels of serum sodium (*p* for trend = 0.005), alpha carotene (*p* for trend = 0.006), and albumin (*p* for trend = 0.005) were associated with a decreased risk of all-cause mortality. In addition, these significant associations were not modified by age, sex, or race. Inclusion of thyroid stimulating hormone (*p* = 0.03), fibrinogen (*p* = 0.01), and urine albumin (*p* < 0.0001), separately, modestly improved the discriminatory ability for predicting all-cause mortality among diabetic patients. Conclusions: Our nationwide study findings provide strong evidence that some nutritional, environmental, and metabolic biomarkers were significant predictors of all-cause mortality among diabetic patients and may have potential clinical value for improving stratification of mortality risk.

## 1. Introduction

In 2015, there were about 30.2 million adults living with diabetes in the US, or 12.2% of all US adults [1]. According to a 2017 National Center for Health Statistics (NCHS) report [2], it is the seventh leading cause of mortality in the US. Patients with diabetes are at increased risk of mortality due to several causes including diabetes itself (for example, diabetic ketoacidosis), cardiovascular disease (CVD) (for example, coronary artery disease), renal disease, cancer, and infections [3]. Therefore, the therapeutic goal is to prevent the progression of diabetes to late stages and related complications [4].

Previous studies [5,6] have investigated nonlaboratory-based risk factors for mortality in diabetic patients and showed that sociodemographic and clinical factors (such as older age, male sex, smoking, lower income, renal disease, macrovascular disease, and longer duration of diabetes) are associated with increased risk of all-cause mortality among diabetic patients with access to medical care [5]. However, diabetes is a complex and heterogeneous disease in terms of clinical and molecular phenotypes. In diabetes, traditional physiological and environmental factors (such as nutrients, sleep, stress, activity, and pollutants) can influence the expression of genes (for example, proteins, metabolites) that can be detected in biospecimens such as blood, urine, or stool [7]. As such, there is increasing interest in identifying prognostic biomarkers among diabetic patients at elevated risk of diabetic complications or mortality, with the ultimate goal of reinforcing preventive strategies [4]. For instance, several studies have investigated predictive biomarkers for diabetic kidney disease, such as estimated glomerular filtration rate (eGFR), albuminuria, inflammatory biomarkers, metabolites, serum creatinine, blood urea, etc. [4,8] or predictive biomarkers for cardiovascular events [9,10,11,12,13,14,15,16,17,18] in the diabetic population.

Previous studies have also investigated the associations between multiple biomarkers and all-cause mortality among diabetic patients. Some of these studies have studied single biomarkers such as red cell distribution width [19], high-sensitive cardiac troponin T [14], copeptin [18], and creatinine excretion rate [20]. Others have focused on multiple biomarkers, such as urinary albumin and nucleic acid oxidation markers in urine [11], urinary markers of nucleic acid oxidation in urine [21], urinary hepatocyte growth factor and adiponectin [12], amino acids [22], the ratio of aspartate aminotransferase (AST) to alanine aminotransferase (ALT) [9], serum sodium, copeptin, and NT-proBNP [15], serum vitamin D deficiency and other factors such as serum albumin among patients on hemodialysis therapy [23] with all-cause mortality in diabetic patients. However, results have been inconsistent and inconclusive. Furthermore, there has been a lack of global mortality risk assessment in the general population. In light of these factors, we suggest that the use of conventional biomarkers can be cost-effective and especially useful for the clinical setting and in public health initiatives.

Here, we specifically aimed to prospectively examine the associations of a panel of nutritional, environmental, and metabolic biomarkers with all-cause mortality among diabetes patients utilizing a nationally representative sample of the US from 1988 to 1994 with linked mortality data.

## 2. Materials and Methods

### 2.1. Study Population

The National Health and Nutrition Examination Survey (NHANES) is a series of cross-sectional, stratified, multistage survey representative of the US non-institutionalized population. NHANES has assessed the health and nutritional status of both adults and children in US for several decades. NHANES I-III were conducted between 1971 and 1994, and current NHANES has continuously collected data in 2-year waves since 1999. Data were collected both through at-home interviews and visits to a mobile examination center (MEC), where participants underwent a physical examination and provided a blood sample. Urine samples were collected either at participants’ homes or the MEC. A standardized questionnaire was given to collect demographic information, such as age, sex, race/ethnicity, and educational level. The study protocol was approved by the National Centers for Health Statistics Ethics Review Board. All participants provided written informed consent.

For this analysis, we used data from 2 survey cycles (1988–1991 and 1991–1994) of the NHANES III. After we excluded subjects younger than 20 years old (*n* = 1225), participants who did not go to the MEC or were examined at home (*n* = 1795), participants without diabetes (*n* = 14,900), women who were pregnant (since pregnancy affects glucose levels) (*n* = 14), and people who did not have mortality status on file (*n* = 3), there were 2113 diabetes patients in the analyses (Appendix A).

### 2.2. Linked Mortality Data through 31 December 2015

Participants from NHANES III (1988–1994) were matched to the National Death Index (NDI) certificate records to determine mortality status through 31 December 2015. The NHANES III linked mortality file is provided by NCHS and the public-use Linked Mortality Files are available for use [24,25].

### 2.3. Demographic Characteristics, Lifestyle and Body Mass Index (BMI)

Participants’ age, race (white, black, others), sex, educational level (less than high school, high school or higher), smoking status, alcohol consumption, and level of physical activity were obtained from self-reports. Smoking and drinking were both categorized as never, former, or current. Never smokers were those who reported smoking no more than 100 cigarettes (approximately 5 packs) during their entire lifetime. Never alcohol drinkers were those who had no more than 12 drinks in their entire life. BMI was defined as weight in kilograms divided by square of height in meters and divided into 2 groups: <25 kg/m^2^ (underweight or normal) and ≥25 kg/m^2^ (overweight or obese) [26].

### 2.4. Physical Activity

Physical activity level was defined according to the frequency and intensity of participant’s self-reported leisure-time physical activity in the past month: inactive, insufficient, and active, based on metabolic equivalent (MET) intensity levels. Participants were defined as active if they were engaged in moderate physical activity (METs 3–6) 20 or more times in the last month, or vigorous physical activity (MET > 6) 12 or more times. Participants were considered inactive if they had no report of leisure-time physical activity. Participants who were not inactive but did not meet the active criteria were classified as insufficiently active [27].

### 2.5. Comorbidities

Comorbidities considered here included self-reported status of cancer, retinopathy, neuropathy, cardiovascular disease (CVD), and chronic kidney disease (CKD). To determine the presence of CKD, estimated glomerular filtration rate (eGFR) was calculated based on the Chronic Kidney Disease Epidemiology Collaboration (CKD-EPI) equation [28]. CKD was further defined as an estimated glomerular filtration rate (eGFR) of <60 mL/minute/1.73 m^2^ [29,30]. A composite comorbidity measure was further included if patients had any of the following comorbidities: cancer, retinopathy, neuropathy, CVD, or CKD.

### 2.6. Laboratory-Based Biomarkers

This study included a panel of 44 biomarkers available from NHANES III. Spot blood and urine specimens were collected and processed under the published detailed laboratory procedures and quality control assessments for NHANES III by Centers for Disease Control and Prevention (CDC) [31].

### 2.7. Diabetes Definition

During the interview, participants were asked if they had ever been told by a doctor that they have diabetes. If the answer was “yes”, we defined it as self-reported previous diagnosis of diabetes after we excluded females who had diabetes when pregnant. In the present study, diabetes was defined if any of the following conditions were met (1) previous diagnosis of diabetes; (2) a hemoglobin A1c (HbA1c) level of 6.5% or greater; (3) fasting plasma glucose (FPG) level of 126 mg/dL or higher; (4) any PG level of 200 mg/dL or higher; (5) use of insulin or diabetic medication.

### 2.8. Statistical Analyses

NHANES oversampled certain groups, such as people older than 60, Mexican Americans, and non-Hispanic black people. The sample weights were used in the analysis, taking into account the complex sampling as well as non-response. Baseline characteristics of each biomarker in weighted means (standard errors) or medians (interquartile ranges) for US adults with diabetes aged 20 years or older, and male and females, separately, are presented in Table 1. Continuous variables of population characteristics are presented as weighted mean (standard error), and categorical variables were presented as weighted percentage (standard error) in Appendix A.

We estimated the relative association between each biomarker and all-cause mortality by categorizing each participant’s biomarker values into tertiles and performing multivariate survey-weighted logistic regression to determine the odds ratios (ORs) and 95% confidence intervals (CIs) for participants in each of the upper 2 tertiles compared to those in the lowest tertile. We adjusted for covariates that were considered potential confounders for all-cause mortality or previously shown to be associated with mortality in diabetic patients [5,6]. A total of 2 models were constructed: Model 1 was adjusted for survey cycle (1988–1991 and 1991–1994), age (continuous), gender, race, and BMI (<25 and ≥25 kg/m^2^); Model 2 was additionally adjusted for smoking status (nonsmoker, past smoker, current smoker), alcohol intake (nondrinker, past drinker, current drinker), physical activity level (inactive, insufficient, active), education level (<high school and ≥high school), and composite comorbidity status (binary). For biomarkers that did not depart from linearity in relation to mortality risk, we then tested for the presence of linear trends by assigning each tertile the median value and modeling this as a continuous variable. Furthermore, we non-parametrically examined the possibility of non-linear relationships between continuous biomarker levels and mortality risk using restricted cubic splines with three knots and adjusted for variables in Model 2.

To assess if the significant associations identified in Model 2 were modified by the effects of age, gender, or race, we performed stratified analysis on these associations based on these potential effect modifiers. By plotting the adjusted ORs (from Model 2) against tertiles of biomarkers stratified by each potential effect modifier, we graphically evaluated potential effect modifications by determining whether the lines were non-parallel [32]. We also derived the Spearman correlation matrix among the significant biomarkers identified from Model 2 to assess cross-correlations between biomarkers.

Prediction analysis was conducted on these significant biomarkers to determine whether the C-statistic is significantly improved by including each of the biomarkers separately to Model 2 using the non-parametric test of Delong et al. [33] comparing the area under two correlated receiver operating characteristic (ROC) curves. All of the analyses were conducted using SAS, version 9.4 (SAS Institute, Inc., Cary, NC, USA). Moreover, 2-sided tests and a 5% significance level were used.

## 3. Results

### 3.1. Patient Characteristics

The study population consisted of 2113 adults aged 20 to 90 years (mean age: 58.2 years) with diabetes. Of these participants, 1172 were women and 955 were men. As of the 2016, 1387 (65.2%) died. The characteristics of participants are shown in Appendix A.

Among the 44 studied biomarkers, the vast majority differed significantly between men and women, with the exception of vitamin E, total carotene, chloride, beta cryptoxanthin, lycopene, bicarbonate, protein, thyroid stimulating hormone (TSH), alkaline phosphatase, sum retinyl esters, and osmolality (Table 1).

### 3.2. Associations of Nutritional, Environmental, and Metabolic Biomarkers with All-Cause Mortality

The results of associations of tertiles of 44 biomarkers with all-cause mortality based on Model 1 and Model 2 are summarized in Table 2. In both models, 9 biomarkers (alpha carotene, sodium, TSH, urine albumin, C-reactive protein [CRP], serum albumin, lactate dehydrogenase [LDH], plasma fibrinogen, and gamma glutamyl transferase [GGT]) were significantly and linearly associated with all-cause mortality.

According to the Model 2 results, higher levels of these 6 biomarkers were significantly associated with increased risk of all-cause mortality: serum CRP (*p* for trend = 0.0004), TSH (*p* for trend = 0.04), LDH (*p* for trend = 0.02), GGT (*p* for trend = 0.02), plasma fibrinogen (*p* for trend = 0.03), and urine albumin (*p* for trend < 0.0001). The highest tertile of CRP was significantly associated with 2.44 times increased risk of all-cause mortality compared to the first tertile. The third tertile of serum TSH was also significantly associated with 70% higher risk of all-cause mortality compared to the first tertile. The third tertiles of LDH, GGT, and plasma fibrinogen were significantly associated with 93%, 67%, and 84% higher risk of all-cause mortality, respectively. The third tertile of urine albumin was associated with 3.72 times increased risk of all-cause mortality.

According to Model 2 results, elevated levels of the following 3 biomarkers were significantly associated with decreased risk of all-cause mortality: serum sodium (*p* for trend = 0.005), alpha carotene (*p* for trend = 0.006), and serum albumin (*p* for trend = 0.005). The third tertiles of serum sodium and alpha carotene were significantly associated with 49% and 57% decreased risk of all-cause mortality, respectively. Although serum albumin showed a significant linear association with decreased all-cause mortality, neither the second tertile (OR 0.56, 95% CI 0.30–1.05) nor the third tertile (OR 0.46, 95% CI 0.28–6.81) was significantly associated with decreased mortality compared to the first tertile.

Among all of these statistically significant relationships, none of the associations was modified by age, race, or sex. Stratified analysis results for serum sodium, alpha carotene, and TSH are shown in Appendix A.

Figure 1 shows the fully adjusted restricted cubic splines models for the associations of multiple biomarkers (significant biomarkers from Model 2 and serum vitamin D) and all-cause mortality. Findings are consistent with Table 2 except urine albumin, whose splines showed a declining pattern in adjusted OR when above 300 µg/dL. 

### 3.3. Cross-Correlation within a Subset of the Significant Biomarkers

Spearman’s correlations between the significant biomarkers identified from Model 2 are shown in Figure 2. The correlation matrix revealed that CRP and LDH were significantly correlated with all other biomarkers. CRP was positively correlated with GGT, urine albumin, LDH, TSH, and fibrinogen, but negatively correlated with alpha carotene, sodium, and serum albumin. CRP showed the strongest correlation with fibrinogen (correlation coefficient = 0.47, *p* < 0.01). LDH was positively correlated with alpha carotene, sodium, GGT, urine albumin, TSH, CRP, and fibrinogen, but negatively correlated with serum albumin. LDH had the strongest correlations with serum albumin (correlation coefficient = −0.14, *p* < 0.01), GGT (correlation coefficient = 0.14, *p* < 0.01), and urine albumin (correlation coefficient = 0.14, *p* < 0.01). In addition, there was also a significant negative correlation between serum and urine albumin (correlation coefficient = −0.27, *p* < 0.01).

### 3.4. Assessing the Values of the Significant Biomarkers for Predicting All-Cause Mortality

Table 3 shows the improvements in C-statistics with the addition of individual biomarkers to Model 2. The C-statistic was modestly but significantly increased with addition of serum TSH (*p* = 0.03), plasma fibrinogen (*p* = 0.01), urine albumin (*p* < 0.0001), individually, or with all these three biomarkers combined (*p* < 0.0001), and with serum TSH and plasma fibrinogen (without urine albumin) (*p* = 0.002).

## 4. Discussion

In a nationally representative sample of US adults with diabetes, we found that higher levels of serum CRP, TSH, LDH, GGT, plasma fibrinogen, and urine albumin were significantly associated with increased risk of all-cause mortality, while higher levels of serum sodium, alpha carotene, and albumin were associated with a decreased risk of all-cause mortality. TSH, plasma fibrinogen, and urine albumin, individually or in combination, significantly improved the prediction of all-cause mortality among diabetic individuals beyond classical risk factors. To our knowledge, this is the first study exploring a comprehensive panel of laboratory-based biomarkers with all-cause mortality among diabetic patients from the general population.

Our finding of the association between higher levels of CRP and increased all-cause mortality is consistent with previous epidemiological studies in either the general population [34] or diabetic patients [35,36,37]. CRP is a protein secreted by the liver that reflects inflammation and is strongly predictive of CVD outcomes, such as with coronary heart disease (CHD), ischemic stroke, vascular mortality, and non-vascular mortality [38] or CVD mortality [37]. Cox et al. [35] studied the association between CRP and all-cause mortality among European Americans with diabetes. They found that those with CRP of 3–10 mg/L and >10 mg/L were at twice and 5 times increased risk of all-cause mortality, respectively [35].

We observed a significant association between higher levels of TSH and increased risk of all-cause mortality. Subclinical hypothyroidism (when thyroid hormones are within the normal range) or overt hypothyroidism (if thyroid hormones are decreased), as reflected by elevated TSH levels, can affect diabetic complications such as risk of CVD and nephropathy [39]. A meta-analysis of 36 studies showed that patients with type 2 diabetes had 1.93 times increased risk of subclinical hypothyroidism compared to normal controls [40]. The same study also reported that subclinical hypothyroidism was significantly associated with increased risk of diabetic complications, including nephropathy, retinopathy, peripheral arterial disease, and peripheral neuropathy [40]. Overall, results from studies of the association between subclinical hypothyroidism and CVD risk and mortality are conflicting [41,42]. Subclinical hypothyroidism has also been associated with albuminuria among diabetic patients [43]. Albuminuria is a biomarker reflecting the severity of diabetic nephropathy [8]. In our study, TSH was positively correlated with urine albumin and both biomarkers were significantly associated with increased risk of mortality. Thus, our findings with TSH and increased risk of all-cause mortality might reflect a subset of patients suffering from diabetic complications (for example, nephropathy) who could have increased risk of mortality, which warrants further studies on cause-specific mortality.

This study also showed that higher levels of plasma fibrinogen were associated with increased all-cause mortality. Fibrinogen is a glycoprotein that is converted to fibrin during coagulation and is a marker of platelet aggregation [44]. Previous epidemiological studies showed that increased fibrinogen levels were associated with increased CVD (for example, CHD and stroke), non-vascular mortality (for example, cancer deaths) [44], or sudden cardiac death [45]. High fibrinogen levels were also associated with increased risk of cardiovascular events among a diabetic population [46]. Our finding of a strong and positive correlation between fibrinogen and CRP is also consistent with previous reports [44] suggesting that they both reflect systemic inflammation whose associations with CVD mortality may explain the associations of fibrinogen and CRP with all-cause mortality among diabetic patients.

To our knowledge, our study is the first study showing that LDH is associated with all-cause mortality among a diabetic population. LDH is an enzyme that reversibly converts lactate to pyruvate [47]. Abnormal levels of LDH have been shown to be correlated with various conditions such as sepsis, malignancies, myocardial infarction (MI), and liver disease [48]. In a retrospective study using NHANES III data, Wu et al. [48] reported that the highest tertile of LDH was significantly associated with all-cause mortality among individuals with metabolic syndrome, which includes diabetes in addition to insulin resistance, hypertension, dyslipidemia, and obesity [48]. Of note, LDH showed significant correlations with other biomarkers (for example, serum albumin, GGT, and urine albumin) in that study. Thus, future studies are needed to confirm its independent association with all-cause mortality.

Our findings indicate an association between higher levels of GGT and all-cause mortality in a diabetic population. GGT is a liver enzyme whose abnormal levels are indicative of hepatic inflammation, non-alcoholic fatty liver disease, type 2 diabetes, and insulin resistance. Epidemiological studies have shown a positive correlation of GGT with BMI, total cholesterol, diabetes, as well as with all-cause and CVD mortality [49]. Previous studies among diabetic patients reported associations between higher levels of GGT and all-cause [50,51], cancer related, and CVD-related mortality [51]; however, a study by Kengne et al. [52] reported that GGT is associated with all-cause and CVD mortality in diabetic and non-diabetic individuals and there was no interaction between diabetes and GGT for all-cause or CVD mortality [52]. We hypothesize that higher GGT levels in this study might indicate a subset of diabetic patients with more cardiometabolic morbidities that could have increased their risk of all-cause mortality.

Consistently, we found that higher levels of urinary albumin were associated with increased risk of all-cause mortality. Microalbuminuria is defined by urine albumin concentrations of 20–200 mg/L; while concentrations above 200 mg/L are considered macroalbuminuria [53]. Previous studies have also shown associations between microalbuminuria and atherosclerosis, CVD, renal disease, and all-cause mortality among diabetic patients [11,54,55]. High urinary albumin may indicate progressive CVD and kidney disease among diabetic patients [56], which could explain the association between microalbuminuria and increased risk of mortality.

We found that higher levels of serum sodium were associated with lower all-cause mortality among diabetic patients. The prevalence of low serum sodium levels (hyponatremia) is higher in patients with diabetes and might indicate disorders involving electrolyte imbalance such as heart failure [15]. In line with our findings, a study among diabetic patients found that lower levels of serum sodium were associated with increased CVD and all-cause mortality; however, the association was irrespective of NT-proBNP (a marker for heart failure) [15].

We also found that higher levels of serum alpha-carotene were associated with decreased risk of all-cause mortality. Alpha-carotene is an antioxidant that is chemically similar to beta-carotene and plays a role in preventing oxidative damage, a process integral to the pathogenesis of chronic diseases such as CVD and cancer [57]. Our finding was consistent with other observational studies showing inverse associations of alpha-carotene with total and cause-specific mortality, such as a study of the general population of NHANES III [57] and a dose-response meta-analysis of 41 prospective cohort studies [58].

Our findings of significant inverse associations between serum albumin and all-cause mortality indicate that patients with higher serum albumin levels had fewer diabetic-related morbidities (for example, diabetic nephropathy). Previous studies have suggested that the synthesis of serum albumin is regulated by insulin levels and that serum albumin levels are inversely associated with glycemic status among diabetic patients [59]. Higher serum albumin levels have been associated with decreased risk of diabetic retinopathy and diabetic kidney disease [60], while hypoalbuminemia (serum albumin level < 3 g/dL) was significantly associated with higher prevalence of ketonuria compared to those with normal albumin levels [59]. Our results of the negative correlation between serum and urine albumin is consistent with another study among diabetic patients showing that serum albumin levels were negatively associated with macroalbuminuria and nephrotic proteinuria [61].

### Strengths and Limitations

This study provided a comprehensive assessment of a panel of 44 biomarkers and their predictive value for all-cause mortality. However, our study has several limitations. First, due to the nature of observational study, causality cannot be inferred for the associations. Second, we relied on one single-time measurement of each biomarker. Although some biomarkers might be more stable over time, the levels of some other biomarkers might change. As such, repeated measurements of biomarkers over time could have strengthened our results. Third, although we adjusted for major confounders, we cannot completely exclude residual confounding especially when we did not adjust for medications or other factors that could have confounded the results.

In conclusion, we found 9 biomarkers out of 44 studied biomarkers to be associated with all-cause mortality among US diabetic patients. Higher levels of serum CRP, TSH, LDH, GGT, plasma fibrinogen, and urine albumin were associated with increased risk of all-cause mortality. On the other hand, higher levels of serum sodium, alpha carotene, and serum albumin were associated with decreased risk of all-cause mortality. Out of these nine biomarkers, TSH, plasma fibrinogen, and urine albumin significantly improved the prediction of all-cause mortality among diabetic patients. These findings warrant further studies to investigate the association of these biomarkers with cause-specific mortalities among diabetic patients.

## Figures and Tables

**Figure 1 nutrients-14-02629-f001:**
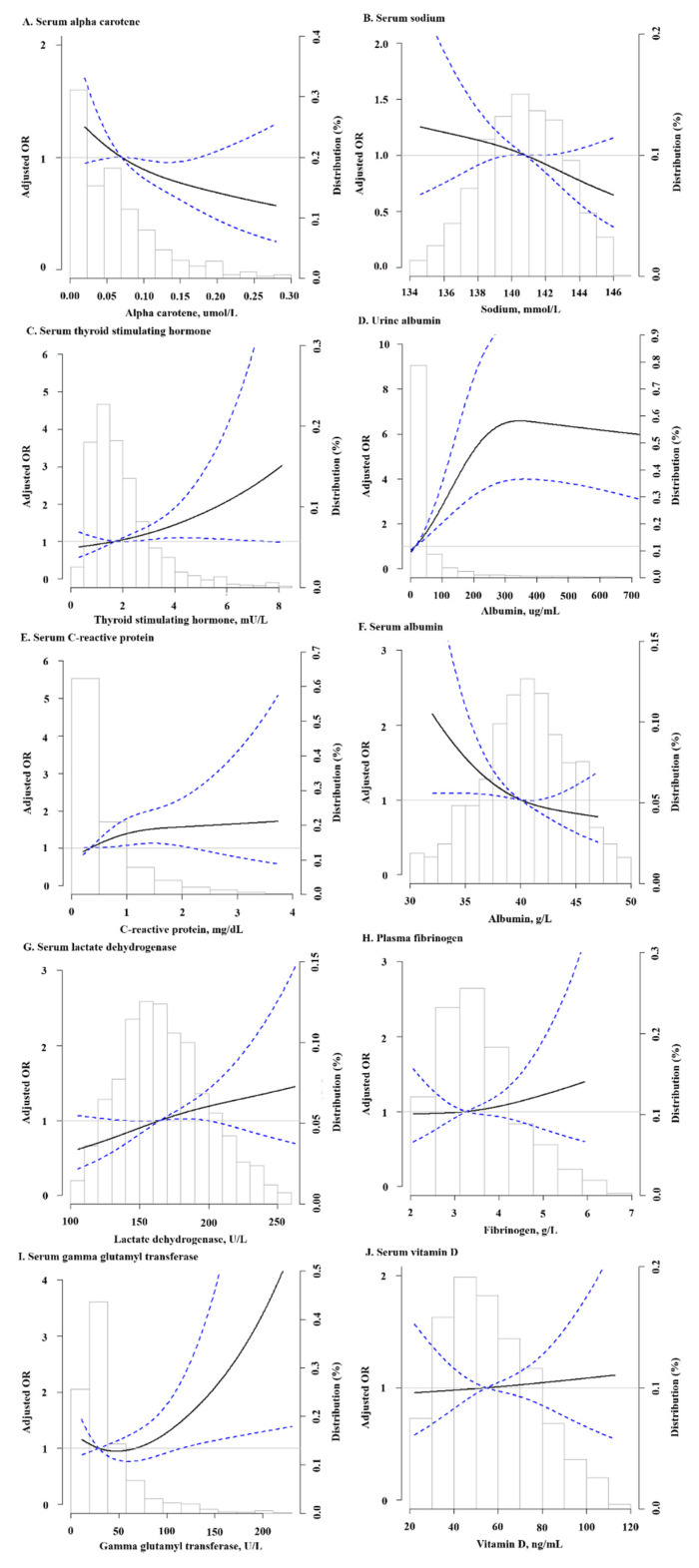
The associations of biomarkers and odds ratio of all-cause mortality using adjusted restricted cubic spline models (95% confidence intervals). (**A**–**J**) showed alpha carotene, serum sodium, se-rum thyroid stimulating hormone, urine albumin, serum C-reactive protein, serum albumin, serum lactate dehydrogenase, plasma fibrinogen, serum gamma glutamyl transferase, and se-rum vitamin D, respectively. The solid line represents the odds ratio and the dashed line repre-sents the 95% confidence intervals. Three knots were specified, whether they were placed auto-matically by program at appropriate percentiles. Splines were obtained where data below 2.5 percentile and above 97.5 percentile were excluded, and median was used as reference value for each biomarker respectively. Histogram shows the distribution of biomarkers. Models were adjusted for survey cycle (binary), age (continuous), gender (binary), race (binary), BMI (contin-uous), smoking status (categorical), alcohol intake (categorical), physical activity (categorical), education level (binary), and composite comorbidity status (binary).

**Figure 2 nutrients-14-02629-f002:**
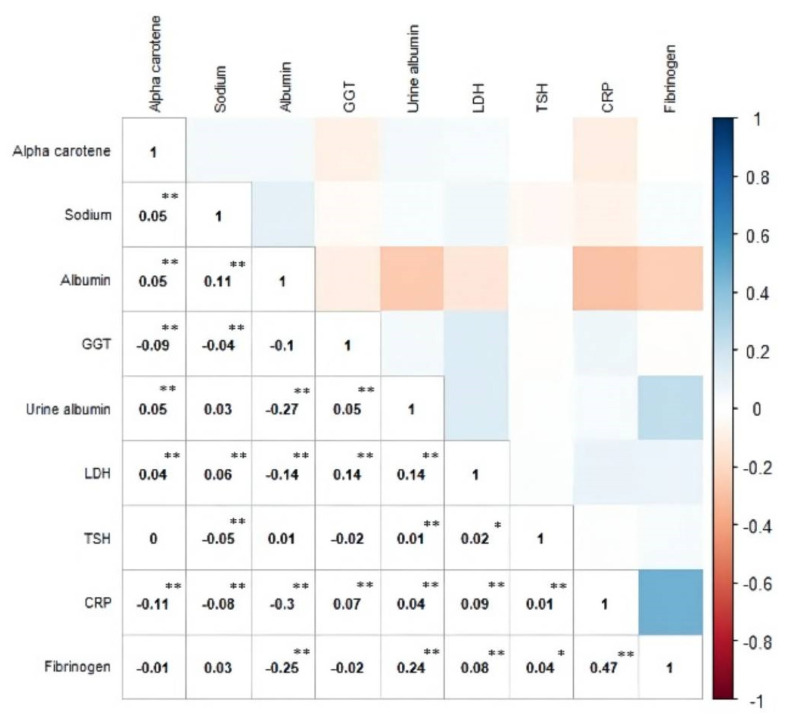
Spearman correlation between significant biomarkers based on Model 2. The figure is ordered by using hierarchical clustering. * denotes the *p*-value for test of correlation coefficient is <0.05; ** denotes the *p*-value is <0.01. TSH: Thyroid stimulating hormone; GGT: gamma glutamyl transferase; CRP: C-creative protein; LDH: lactate dehydrogenase. All biomarkers are from blood samples except urine albumin.

**Table 1 nutrients-14-02629-t001:** Baseline biomarker characteristics of US adults with diabetes aged over 20 years in NHANES III *.

Biomarkers	All	Men	Women	*p*-Value
Participants (*n*)	2113	955	1158	
**Nutritional biomarkers**				
**Blood**				
Vitamin A (μmol/L)	2.11 (0.73)	2.21 (0.77)	2.02 (0.68)	<0.0001
Vitamin C (mmol/L)	36.5 (23.2)	33.6 (22.5)	38.9 (23.4)	<0.0001
Vitamin D (nmol/L)	57.6 (24.6)	62.1 (23.5)	53.9 (24.9)	<0.0001
Vitamin E (μmol/L)	30.2 (15.6)	29.5 (15.1)	30.9 (16.0)	0.06
Alpha carotene (μmol/L)	0.07 (0.04–0.11)	0.07 (0.04–0.09)	0.07 (0.04–0.11)	0.002
Beta carotene (μmol/L)	0.28 (0.17–0.47)	0.26 (0.15–0.43)	0.32 (0.17–0.50)	<0.0001
Total carotene (μmol/L)	1.29 (0.93–1.75)	1.28 (0.90–1.74)	1.30 (0.96–1.77)	0.21
Iron (μmol/L)	14.3 (6.13)	15.7 (6.53)	13.1 (5.51)	<0.0001
Ferritin (ug/L)	138 (67–259)	181 (102–330)	105 (48.0–201)	<0.0001
Total iron binding capacity (μmol/L)	61.9 (10.7)	60.4 (9.92)	63.2 (11.1)	<0.0001
Calcium (mmol/L)	2.30 (0.13)	2.29 (0.13)	2.31 (0.13)	0.001
Selenium (nmol/L)	1.58 (0.24)	1.60 (0.26)	1.57 (0.22)	0.004
Sodium (mmol/L)	141 (3.10)	141 (2.90)	140 (3.25)	0.04
Chloride (mmol/L)	103 (3.96)	103 (3.82)	103 (4.07)	0.16
Potassium (mmol/L)	4.08 (0.41)	4.14 (0.42)	4.03 (0.40)	<0.0001
Phosphorus (mmol/L)	1.11 (0.19)	1.08 (0.20)	1.13 (1.03–1.26)	<0.0001
Lutein/zeaxanthin (μmol/L)	0.37 (0.26–0.53)	0.40 (0.28–0.54)	0.35 (0.26–0.53)	0.005
Beta cryptoxanthin (μmol/L)	0.19 (0.16)	0.19 (0.16)	0.20 (0.16)	0.10
Lycopene (μmol/L)	0.34 (0.21)	0.34 (0.22)	0.34 (0.21)	0.77
Globulin (g/L)	34.7 (5.12)	33.9 (5.19)	35.2 (4.99)	<0.0001
Bicarbonate (mmol/L)	31.6 (14.4)	31.4 (13.7)	31.7 (14.9)	0.69
Folate (nmol/L)	16.4 (14.3)	15.7 (12.4)	17.1 (15.8)	0.03
Protein (g/L)	74.1 (5.23)	73.9 (5.25)	74.2 (5.22)	0.33
**Urine**				
Creatinine (mmol/L)	10.2 (6.17)	11.7 (6.18)	8.90 (5.87)	<0.0001
Albumin (ug/mL)	14.8 (5.50–50.0)	16.8 (6.30–60.5)	13.4 (4.95–45.0)	0.0009
Iodine (ug/dL)	15.0 (8.80–23.2)	15.5 (9.80–24.2)	14.3 (7.80–22.1)	0.0002
**Environmental Biomarkers**				
**Blood**				
Lead (μmol/L)	0.16 (0.11–0.26)	0.21 (0.14–0.32)	0.14 (0.09–0.20)	<0.0001
Cotinine (ng/mL)	0.27 (0.08–10.1)	0.38 (0.10–78.9)	0.22 (0.07–2.02)	<0.0001
**Urine**				
Cadmium (nmol/L)	5.16 (2.58–9.61)	5.34 (2.94–10.05)	5.07 (2.40–9.16)	0.009
**Metabolic biomarkers**				
**Blood**				
Creatinine (μmol/L)	97.2 (79.6–115)	106 (97.2–124)	88.4 (79.6–97.2)	<0.0001
C-reactive protein (mg/dL)	0.33 (0.21–0.80)	0.21 (0.21–0.66)	0.40 (0.21–1.00)	<0.0001
Urea nitrogen (mmol/L)	6.22 (3.21)	6.47 (3.14)	6.02 (3.26)	0.002
Uric acid (μmol/L)	335 (102)	356 (100)	317 (100)	<0.0001
Albumin (g/L)	39.9 (4.01)	40.5 (4.19)	39.5 (3.78)	<0.0001
Lactate dehydrogenase (U/L)	171 (41.3)	167.9 (41.6)	173 (41.0)	0.004
Alkaline phosphatase (U/L)	104 (45.8)	102 (49.6)	105 (42.3)	0.21
Sum retinyl esters (μmol/L)	0.17 (0.10–0.24)	0.17 (0.10–0.24)	0.17 (0.10–0.24)	0.76
Bilirubin (μmol/L)	9.77 (5.44)	11.2 (6.13)	8.60 (4.45)	<0.0001
Osmolality (mmol/Kg)	284 (7.62)	284 (7.40)	284 (7.80)	0.51
Fibrinogen (g/L)	3.47 (1.03)	3.41 (1.06)	3.51 (1.01)	0.047
Gamma glutamyl transferase (U/L)	29.0 (20.0–46.0)	31.0 (22.0–53.0)	28.0 (19.0–43.0)	<0.0001
Aspartate aminotransferase (U/L)	19.0 (15.0–24.0)	20.0 (16.5–25.0)	18.0 (15.0–23.0)	<0.0001
Alanine aminotransferase (U/L)	15.0 (10.0–21.0)	17.0 (12.0–24.0)	14.0 (10.0–20.0)	<0.0001
Thyroid stimulating hormone (mU/L)	1.70 (1.10–2.60)	1.70 (1.10–2.50)	1.72 (1.10–2.70)	0.37

* Sample sizes are unweighted. Survey weight-adjusted means (standard error) and median (Q1–Q3) are presented. *p*-values were based on t test and Wilcoxon rank-sum test between men and women.

**Table 2 nutrients-14-02629-t002:** Multivariable-adjusted ORs and 95% CIs of all-cause mortality by tertiles of different biomarkers *.

	ORs (95% CI) of Model 1		*p* for Linear Trend ^†^	ORs (95% CI) of Model 2		*p* for Linear Trend ^†^
	Tertile 2 ^‡^	Tertile 3		Tertile 2	Tertile 3	
**Nutritional biomarkers**						
**Blood**						
Vitamin A (μmol/L)	0.61 (0.37–1.001)	1.15 (0.66–1.98)	0.35	0.69 (0.41–1.18)	1.32 (0.74–2.35)	0.17
Vitamin C (mmol/L)	0.97 (0.54–1.76)	0.68 (0.40–1.16)	0.13	1.27 (0.73–2.23)	0.95 (0.56–1.61)	0.75
Vitamin D (nmol/L)	0.70 (0.49–1.002)	0.83 (0.52–1.34)	0.66	0.82 (0.54–1.23)	0.92 (0.56–1.53)	0.90
Vitamin E (μmol/L)	1.09 (0.72–1.66)	1.24 (0.77–1.98)	0.40	1.03 (0.65–1.62)	1.46 (0.91–2.34)	0.09
Alpha carotene (μmol/L)	0.78 (0.48–1.27)	0.37 (0.25–0.57)	<0.0001	0.94 (0.56–1.59)	0.44 (0.26–0.74)	0.006
Beta carotene (μmol/L)	0.65 (0.43–0.99)	0.59 (0.39–0.91)	0.03	0.76 (0.46–1.27)	0.67 (0.41–1.10)	0.11
Total carotene (μmol/L)	0.60 (0.39–0.93)	0.62 (0.35–1.09)	0.11	0.68 (0.42–1.10)	0.72 (0.39–1.34)	0.33
Iron (μmol/L)	0.60 (0.39–0.91)	0.74 (0.48–1.14)	0.25	0.60 (0.41–0.88)	0.83 (0.53–1.31)	0.71
Ferritin (ug/L)	0.57 (0.34–0.94)	1.03 (0.62–1.71)	0.26	0.65 (0.37–1.13)	1.13 (0.61–2.06)	0.26
Total iron binding capacity (μmol/L)	0.81 (0.52–1.27)	0.99 (0.64–1.53)	0.94	0.77 (0.49–1.22)	0.99 (0.66–1.50)	0.87
Calcium (mmol/L)	0.70 (0.47–1.06)	1.10 (0.65–1.86)	0.94	0.76 (0.50–1.17)	0.98 (0.59–1.64)	0.77
Selenium (nmol/L)	0.54 (0.35–0.85)	0.60 (0.34–1.05)	0.12	0.64 (0.41–0.996)	0.77 (0.43–1.38)	0.52
Sodium (mmol/L)	0.84 (0.57–1.23)	0.53 (0.36–0.80)	0.004	0.92 (0.61–1.40)	0.51 (0.33–0.78)	0.005
Chloride (mmol/L)	0.77 (0.49–1.23)	0.89 (0.50–1.57)	0.64	0.81 (0.50–1.33)	0.75 (0.43–1.31)	0.30
Potassium (mmol/L)	1.34 (0.79–2.28)	1.19 (0.72–1.97)	0.53	1.33 (0.80–2.20)	1.15 (0.72–1.84)	0.54
Phosphorus (mmol/L)	0.99 (0.60–1.65)	1.21 (0.78–1.89)	0.43	0.91 (0.53–1.56)	1.02 (0.63–1.64)	0.99
Lutein/zeaxanthin (μmol/L)	0.73 (0.47–1.12)	0.88 (0.54–1.47)	0.70	0.72 (0.45–1.16)	1.01 (0.61–1.68)	0.84
Beta cryptoxanthin (μmol/L)	0.88 (0.55–1.42)	0.79 (0.54–1.16)	0.22	0.98 (0.59–1.63)	1.01 (0.67–1.52)	0.96
Lycopene (μmol/L)	0.59 (0.35–1.00)	0.69 (0.42–1.13)	0.24	0.65 (0.36–1.16)	0.77 (0.46–1.29)	0.49
Globulin (g/L)	1.19 (0.75–1.89)	1.12 (0.77–1.64)	0.48	1.11 (0.71–1.74)	1.26 (0.84–1.91)	0.25
Bicarbonate (mmol/L)	0.90 (0.56–1.43)	1.01 (0.61–1.69)	0.94	0.88 (0.52–1.49)	1.10 (0.65–1.87)	0.77
Folate (nmol/L)	0.74 (0.45–1.22)	0.67 (0.47–0.96)	0.051	0.80 (0.47–1.37)	0.81 (0.55–1.21)	0.38
Protein (g/L)	0.69 (0.42–1.11)	0.74 (0.43–1.27)	0.20	0.72 (0.43–1.19)	0.79 (0.48–1.31)	0.28
**Urine**						
Cadmium (nmol/L)	1.22 (0.75–1.98)	1.58 (0.97–2.58)	0.08	1.21 (0.66–1.89)	1.23 (0.72–2.10)	0.48
Creatinine (mmol/L)	1.07 (0.69–1.66)	0.97 (0.60–1.55)	0.86	1.06 (0.64–1.76)	0.81 (0.48–1.36)	0.37
Albumin (ug/mL)	1.13 (0.70–1.85)	4.08 (2.40–6.92)	<0.0001	1.10 (0.66–1.83)	3.75 (2.17–6.49)	<0.0001
Iodine (ug/dL)	1.32 (0.86–2.04)	1.35 (0.77–2.38)	0.33	1.33 (0.87–2.05)	1.19 (0.67–2.09)	0.62
**Environmental Biomarkers**						
**Blood**						
Lead (μmol/L)	1.38 (0.86–2.21)	1.55 (0.94–2.56)	0.09	1.33 (0.82–2.17)	1.33 (0.80–2.21)	0.28
Cotinine (ng/mL)	1.12 (0.66–1.86)	2.02 (1.23–3.31)	0.02	1.12 (0.64–1.95)	1.07 (0.54–2.12)	0.99
**Urine**						
Cadmium (nmol/L)	1.22 (0.75–1.98)	1.58 (0.97–2.58)	0.08	1.21 (0.66–1.89)	1.23 (0.72–2.10)	0.48
**Metabolic biomarkers**						
**Blood**						
Creatinine (μmol/L)	1.20 (0.71–2.02)	1.51 (0.81–2.83)	0.20	1.23 (0.74–2.07)	1.33 (0.69–2.59)	0.37
C-reactive protein (mg/dL)	1.10 (0.71–1.70)	2.60 (1.60–4.24)	0.0001	0.97 (0.61–1.55)	2.36 (1.42–3.94)	0.0004
Urea nitrogen (mmol/L)	1.05 (0.65–1.69)	1.47 (0.74–2.89)	0.26	1.18 (0.68–2.08)	1.34 (0.68–2.62)	0.40
Uric acid (μmol/L)	0.90 (0.54–1.49)	1.42 (0.90–2.23)	0.11	0.92 (0.57–1.50)	1.42 (0.88–2.30)	0.13
Albumin (g/L)	0.52 (0.29–0.94)	0.41 (0.24–0.69)	0.001	0.56 (0.29–1.05)	0.43 (0.24–0.76)	0.005
Lactate dehydrogenase (U/L)	1.38 (0.91–2.09)	2.04 (1.23–3.39)	0.006	1.36 (0.87–2.13)	1.94 (1.11–3.39)	0.02
Alkaline phosphatase (U/L)	1.64 (1.01–2.64)	1.57 (1.06–2.33)	0.04	1.77 (1.11–2.84)	1.38 (0.89–2.14)	0.08
Sum retinyl esters (μmol/L)	1.26 (0.75–2.11)	1.22 (0.74–2.03)	0.45	1.34 (0.77–2.31)	1.25 (0.74–2.11)	0.41
Bilirubin (μmol/L)	0.93 (0.55–1.57)	0.86 (0.46–1.61)	0.65	1.004 (0.58–1.75)	1.07 (0.57–2.01)	0.82
Osmolality (mmol/Kg)	0.68 (0.48–0.96)	1.12 (0.64–1.95)	0.83	0.71 (0.51–0.98)	1.03 (0.55–1.92)	0.77
Fibrinogen (g/L)	1.08 (0.72–1.61)	2.36 (1.46–3.82)	0.0006	1.02 (0.67–1.54)	1.84 (1.08–3.14)	0.03
Gamma glutamyl transferase (U/L)	1.30 (0.82–2.05)	1.74 (1.05–2.89)	0.03	1.27 (0.82–1.97)	1.77 (1.09–2.88)	0.02
Aspartate aminotransferase (U/L)	0.71 (0.47–1.08)	0.84 (0.53–1.34)	0.63	0.78 (0.52–1.17)	1.003 (0.63–1.59)	0.91
Alanine aminotransferase (U/L)	0.62 (0.40–0.96)	0.66 (0.41–1.04)	0.21	0.67 (0.44–1.02)	0.75 (0.48–1.16)	0.36
Thyroid stimulating hormone (mU/L)	0.86 (0.56–1.33)	1.75 (1.09–2.80)	0.01	0.78 (0.52–1.18)	1.61 (0.97–2.67)	0.04

* All models were constructed by multivariate survey-weighted logistic regression model. Model 1 was adjusted for survey cycle (1988–1991, 1991–1994), age (continuous), gender (binary), race (whites, non-whites), and BMI (<25 and ≥25). Model 2 was additionally adjusted for smoking status (nonsmoker, past smoker, current smoker), alcohol intake (nondrinker, past drinker, current drinker), physical activity (inactive, insufficient, active), education level (<high school, ≥high school), and composite comorbidity status (binary). Comorbidity is defined if the patient has any of the following diseases: cancer, retinopathy, neuropathy, cardiovascular diseases, or chronic kidney disease. ^†^ Medians of biomarkers in each tertile were used to create a continuous variable for the test on linear trend. ^‡^ Tertile-1 was the reference group.

**Table 3 nutrients-14-02629-t003:** Improvements in predictive values with the addition of different biomarkers to model.

Biomarkers	C Statistic in Model without Biomarker *	C Statistic in Model with Biomarker as Tertiles *	Change in C Statistic (95% CI)	*p*-Value
**Blood**				
Alpha carotene	0.739	0.739	−0.0005 (−0.0016, 0.0017)	0.96
Sodium	0.738	0.738	0.0003 (−0.0011, 0.0016)	0.68
Thyroid stimulating hormone	0.737	0.746	0.0086 (0.0011, 0.0161)	0.03
C-reactive protein	0.738	0.741	0.0022 (−0.0013, 0.0058)	0.22
Albumin	0.738	0.742	0.0040 (−0.0018, 0.0098)	0.17
Lactate dehydrogenase	0.738	0.741	0.0027 (−0.0021, 0.0075)	0.26
Fibrinogen	0.722	0.735	0.0128 (0.0030, 0.0229)	0.01
Gamma glutamyl transferase	0.743	0.746	0.0029 (−0.0019, 0.0076)	0.24
**Urine**				
Albumin	0.735	0.761	0.0265 (0.0143, 0.0387)	<0.0001
Blood thyroid stimulating hormone and fibrinogen	0.722	0.741	0.0190 (0.0068, 0.0313)	0.002
Blood thyroid stimulating hormone, fibrinogen, and urine albumin	0.719	0.762	0.0432 (0.0255, 0.0607)	<0.0001

* Models were adjusted for age (continuous), gender (binary), race (white, non-white), BMI (<25, ≥25), smoking status (nonsmoker, past smoker, current smoker), alcohol intake (nondrinker, past drinker, current drinker), physical activity (inactive, insufficient, active), education level (<high school, ≥high school), and composite comorbidity status (binary).

## Data Availability

National Health and Nutrition Examination Survey data underlying this article can be accessed at NHANES—National Health and Nutrition Examination Survey Homepage (cdc.gov accessed on 5 March 2020).

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
