# Peer review of "Associations of Nutritional, Environmental, and Metabolic Biomarkers with Diabetes-Related Mortality in U.S. Adults: The Third National Health and Nutrition Examination Surveys between 1988–1994 and 2016"

_nutrients, 2022, doi:10.3390/nu14132629_

Round 1

Reviewer 1 Report

Dear Authors,

Thank you for your manuscript. The paper presents data from a comprehensive and well-designed study aiming to test the associations between 44 biomarkers of different origins and all-cause mortality risk in the U.S.diabetes patients.

I have only a few comments for your consideration.

First, I suggest providing a flow chat on study participants' recruitment as there were multiple stages.

Next, I suggest better organizing your Methods section 2.3 and adding lifestyle factors to its title. I suggest starting each measure in a separate paragraph: sociodemographic characteristics, body mass index, smoking, alcohol consumption, physical activity, and comorbidities.

Please provide the age range at the beginning of section 3.1. as it is given only in the abstract. Also, the abstract should present the mean (SD) age of the study population.

In the footnotes of Table 2, it would be helpful to explain the meaning of T2 and T3 (tertile-?) and to indicate the reference group (T1-?). 

Probably due to my poor eyesight, I can not read axes' names and numbers in Figure 1. Can these graphs be enlarged and have a better resolution?

Author Response

1. Thank you for your manuscript. The paper presents data from a comprehensive and well-designed study aiming to test the associations between 44 biomarkers of different origins and all-cause mortality risk in the U.S. diabetes patients. I have only a few comments for your consideration.

RE: Thank you very much for your positive comments and kind words. According to your comments we have revised our manuscript.

2. First, I suggest providing a flow chat on study participants' recruitment as there were multiple stages.

RE: Thank you for your suggestion. We have provided a flow chart for participants’ selection in Supplemental figure 1.

3. Next, I suggest better organizing your Methods section 2.3 and adding lifestyle factors to its title. I suggest starting each measure in a separate paragraph: sociodemographic characteristics, body mass index, smoking, alcohol consumption, physical activity, and comorbidities.

RE: Thank you for your suggestion. Accordingly, we have reconstructed these paragraphs and provided a separate title for each of these characteristics.

4. Please provide the age range at the beginning of section 3.1. as it is given only in the abstract. Also, the abstract should present the mean (SD) age of the study population.

RE: Per your suggestion, we have provided both range and mean (SD) for age in the abstract and results section.

5. In the footnotes of Table 2, it would be helpful to explain the meaning of T2 and T3 (tertile-?) and to indicate the reference group (T1-?). 

RE: The meaning of T2 and T3, as well as the reference group (T1) have been added as the footnote in table 2. Thank you!

6. Probably due to my poor eyesight, I can not read axes' names and numbers in Figure 1. Can these graphs be enlarged and have a better resolution?

RE: We have re-plotted Figure 1 in a high resolution. Thank you for pointing this out.

Reviewer 2 Report

This original manuscript focuses on evaluating the association between nutritional, environmental, and metabolic status and the progression and prognosis of type 2 diabetes all-cause mortality among diabetic patients

This is an interesting article. However, in my opinion the paper has major shortcomings that should be addressed.

Tables and figures:

Figure 1, 2 and 3. It is necessary to improve quality tables to make it more attractive and clearer

Table 1. The name of the biomarkers should be aligned to left  margin

Figure 1: Please make the letters bigger in the axis labels. It is difficult to read them even zooming them.

What TIBC stands for? Total Iron Binding Capacity? Meaning of acronyms must appear within the text and/or the caption of each table/figure.

Table 3: blood biomarkers should be aligned to the left margin or centered.

The English language must be revised along the manuscript, as it is not accurate, and typo errors must be corrected as well.

Orthography:

Line 27: “To evaluate the associations between nutritional, environmental, and metabolic status and the progression and progno-28 sis of type 2 diabetes all-cause mortality among diabetic patients.” Some connector is missing at the beginning, such as: “Aim” or “Objective”.

Author Response

1. This original manuscript focuses on evaluating the association between nutritional, environmental, and metabolic status and the progression and prognosis of type 2 diabetes all-cause mortality among diabetic patients. This is an interesting article. However, in my opinion the paper has major shortcomings that should be addressed.

RE: Thanks a lot for all your comments and suggestions. All these shortcomings you ponited have been well considered and revised.

2. Tables and figures: Figure 1, 2 and 3. It is necessary to improve quality tables to make it more attractive and clearer

RE: Thank you very much. We have re-organized all tables and improved the resolution of the figures.

3. Table 1. The name of the biomarkers should be aligned to left margin

RE: Per your suggestion, these biomarkers’ names have been aligned to the left margin in all tables.

4. Figure 1: Please make the letters bigger in the axis labels. It is difficult to read them even zooming them.

RE: Thank you. This has been well taken.

5. What TIBC stands for? Total Iron Binding Capacity? Meaning of acronyms must appear within the text and/or the caption of each table/figure.

RE: TIBC stands for total iron binding capacity. We have revised and provided the full name of TIBC in all tables. Thank you!

6. Table 3: blood biomarkers should be aligned to the left margin or centered.

RE: Thank you. These biomarkers’ names have been aligned to the left margin in all tables.

7. The English language must be revised along the manuscript, as it is not accurate, and typo errors must be corrected as well.

RE: All authors have made a proof reading for the new version of the manuscript to carefully avoiding the typo and language issues.

8. Orthography: Line 27: “To evaluate the associations between nutritional, environmental, and metabolic status and the progression and progno-28 sis of type 2 diabetes all-cause mortality among diabetic patients.” Some connector is missing at the beginning, such as: “Aim” or “Objective”.

RE: We have rewritten this sentence as “We aimed to evaluate the associations of nutritional, environmental, and metabolic status with disease progression and prognosis among diabetic patients”.